

# Laboratory modelling of urban flooding: strengths and challenges of distorted scale models

Xuefang Li[1], Sébastien Erpicum[1], Martin Bruwier[1], Emmanuel Mignot[2], Pascal Finaud-Guyot[3], Pierre Archambeau[1], Michel Pirotton[1], Benjamin Dewals[1]

[1] Hydraulics in Environmental and Civil Engineering (HECE), University of Liège (ULiège), Liège, 4000, Belgium

[2] LMFA, CNRS-Université de Lyon, INSA de Lyon, Lyon, 69100, France

[3] ICube laboratory (UMR 7357), Fluid mechanics team, ENGEES, Strasbourg, France

*Correspondence to*: Xuefang Li (xuefang.li@uliege.be)

**Abstract**

Laboratory experiments are a viable approach to improve process-understanding and generate data for the validation of computational models. However, laboratory scale models of urban flooding are often distorted, i.e. different scale factors are used in the horizontal and vertical directions. This may result in artefacts when transposing the laboratory observations to the prototype scale. The magnitude of such artefacts was not studied in the past for the specific case of urban flooding. Here, we present a preliminary assessment of these artefacts based on the reanalysis of two recent experimental datasets related to flooding of a group of buildings and of an entire urban district, respectively. The results reveal that, in the tested

configurations, the influence of model distortion on the upscaled values of water depths and discharges are both of the order of 10 %. This is comparable to other sources of uncertainties, such as on hydrological data.

**Keywords**

Flood risk; laboratory model; scale factor; distortion effects; urban flooding

## 1. Introduction

Worldwide, floods are the most frequent natural disasters and they cause over one third of overall economic losses due to natural hazards (UNISDR, 2015). Flood losses are particularly severe in urban environments and urban flood risk is expected to further increase over the 21st century (Chen et al., 2015; Lehmann et al., 2015; Mallakpour and Villarini,

2015). In response, concepts such as *water-sensitive urban design* and the *sponge city* model are rapidly developing (Gaines, 2016; Liu, 2016). However, the design and sizing of measures aiming at enhancing urban flood protection require accurate tools for risk modelling and scenario analysis (Wright, 2014).

Specifically, reliable predictions of flood *hazard* are a prerequisite to support flood risk management policies. This includes the accurate estimation of inundation extent, spatial distribution of water depth, discharge partition and flow

velocity in urbanized flood prone areas, since these parameters are critical inputs for flood impact modelling (Dottori et al., 2016; Kreibich et al., 2014; Molinari and Scorzini, 2017). State-of-the-art numerical inundation models benefit from increasingly available remote sensing data, such as laser altimetry. Nonetheless, their validation for urban flood configurations remains incomplete because reference data from the field are relatively scarce and, to a great extent, inadequate (Dottori et al., 2013). Mostly watermarks and aerial imagery are available; but they remain uncertain and

insufficient to reflect the whole complexity of inundation flows, especially in densely urbanized floodplains. Additional information on the velocity fields and discharge partitions are necessary to understand the multi-directional flow



pathways induced by the built-up network of streets and open areas, buildings and underground systems (such as the drainage network, Rubinato et al., 2017), particularly under more extreme flood conditions than ever observed in the past. When available, pointwise velocity measurements remain limited in number due to the challenging conditions for field measurement during a major urban flood event (Brown and Chanson, 2012, 2013).

To complement field data, laboratory models may be a viable alternative, since they enable accurate measurements of flow characteristics under controlled conditions. Laboratory scale models consist in replicating a full-scale configuration (also called *prototype*), at a smaller scale. The *scale factor* of such a model is defined as the ratio $L_p / L_m$ between a characteristic length $L_p$ in the prototype and the corresponding characteristic length $L_m$ in the model (Novak, 1984). The design of scale models is based on the similarity theory, which states that the ratio between the dominating forces

governing the flow should remain the same in the model and in the prototype. For free surface flows, Froude similarity is generally used: the ratio between inertia forces and the gravity are kept identical at the both scales. This implies that the Froude number $F_m$ in the model remains the same as the Froude number $F_p$ in the prototype, where the Froude number is defined as $F = V / ( g H )^{0.5}$, with $V$ a characteristic flow velocity (ms$^{-1}$), $g$ the gravity acceleration (ms$^{-2}$) and $H$ a characteristic water depth (m).

Caution must be taken when interpreting observations from laboratory models because not all the ratios of forces can be kept the same in the prototype and in the model. For instance, when the Froude similarity is applied, the ratio between inertia and viscous forces may vary between the prototype and the model. This may result in so-called *scale effects*, which are artefacts arising from the reduced size of the model compared to the real-world configuration. The ratio between inertia and viscous forces is expressed through the Reynolds number: $R = 4 R_H V / \nu$, with $R_H$ a characteristic value of

the hydraulic radius (ratio between the flow area and the wetted perimeter, in m) and $\nu$ the kinematic viscosity of water (m$^2$s$^{-1}$). According to Froude similarity and without distortion (see Section 2.1), $R$ scales with the power $- 3 / 2$ of the scale factor of the model. Therefore, the magnitude of the scale effects tends to be magnified when a larger scale factor is used. Still, for large enough Reynolds numbers (i.e. sufficiently turbulent flow) both on the prototype and the model, the impact of the scale effects remains limited. For free surface flow, Chanson (2004) recommends to keep the Reynolds

number above 5,000 for the lowest flow rate, and to use scale factors below 25 to 50; but these values were never ascertained for urban flooding models.

Recently, urban flooding has been analysed experimentally in laboratory setups that aim at representing inundation flow across a whole urban district. To cover an entire urban district in a laboratory, while keeping the Reynolds number reasonably high and limiting the measurement errors, distinct scale factors have been used along the horizontal and the

vertical directions, leading to so-called *distorted* scale models. This type of models was used previously for various applications (e.g., fluvial morphodynamics); but its use for urban flooding studies is relatively new (Finaud-Guyot et al., 2018; Güney et al., 2014; Smith et al., 2016). While scale effects in general were investigated in the past for a range of configurations, such as the experimental representation of impulse waves (Heller et al., 2008) or the hydraulics of piano key weirs (Erpicum et al., 2016), the specific artefacts arising from the model distortion were hardly studied and,

particularly, not for experimental models of urban flooding. Given the overwhelming importance of urban flooding, the present paper focuses specifically on the effects of model distortion in laboratory modelling of urban flooding at the district level. It presents a reanalysis of two recent experimental datasets (Araud. 2012; Velickovic et al., 2017) to find out to which extent a strong distortion of an entire urban district model affects the observed water depths and flow partition in-between the streets.





Section 2 provides background information and details the motivations of the study. The considered datasets and the methodology are described in section 3. The results are presented in section 4 and implications thereof are discussed in section 5. Finally, conclusions are drawn in section 6.

## 2. Background and motivation

### 2.1 Undistorted and distorted models of urban flooding

Laboratory models were used for studying urban flooding at various levels, ranging from limited spatial extents (e.g., a single storm drain) up to the level of a whole urban district. Some laboratory models focusing on a limited spatial extent were constructed at the prototype scale, i.e. with a scale factor equal to unity (Djordjevic et al., 2013; Lopes et al., 2013, 2017). At intermediate levels, such as when a single street or a single crossroad is represented, scale factors of the order of 10 to 20 were used (Lee et al., 2013; Mignot et al., 2013; Rivière et al., 2011) which is generally deemed not to lead to excessive scale effects (Chanson, 2004). In contrast, when it comes to the experimental analysis of flooding at the level of an entire urban district, the spatial extent of the prototype to be represented becomes considerably larger ($\sim 10^2 \div 10^3$ m), as summarized in Table 1 and sketched in Supplement 1, so that scale factors reach values as high as $100 \div 200$.

LaRocque et al. (2013) focused on a limited portion of an urbanized floodplain (280 m ×124 m) and used a scale factor of 50, leading to a model Reynolds number of the order of $\sim 7 \times 10^4$. Testa et al. (2007) considered a scale factor of 100 to study transient flooding of a group of buildings extending over 160 m $\times$ 120 m. This led nonetheless to model Reynolds numbers exceeding $10^5$ because extreme flooding scenarios were tested (dam break-induced flood). To analyse river flooding at the level of an entire urban district (1 km $\times$ 2 km), Ishigaki et al. (2003) also used a scale factor of 100; but in this case, despite a particularly large experimental facility (10 m $\times$ 20 m), the model Reynolds number was of the order of $7 \times 10^3$ with water depth lower than 1 cm, and even below in some streets. Ishigaki et al. (2003) reported that the observed flow became "laminar" in some parts of the model, hence exacerbating scale effects. This questions the validity of the upscaled lab observations, and highlights a difficulty in the design of experimental models for analysing urban flooding, namely the substantial difference between the characteristic length in the horizontal direction (e.g. street width) and in the vertical direction (e.g. water depth). This issue is similar to the case of laboratory models of large rivers and coastal systems (Wakhlu, 1984).

To partly overcome this difficulty, so-called *distorted* laboratory models have also been used. They consist in applying distinct scale factors, respectively, $e_H$ and $e_V$, along the horizontal and vertical directions:

$$e_H = \frac{L_p}{L_m} \qquad \text{and} \qquad e_V = \frac{H_p}{H_m}, \qquad (1)$$

where $H_p$ and $H_m$ are characteristic heights in the prototype and in the model. Since $L_p$ is always much higher than $H_p$, using a horizontal scale factor $e_H$ larger than the vertical scale factor $e_V$ enables preserving higher water depths in the laboratory model compared to an equivalent undistorted model requiring the same horizontal space in the laboratory. This approach offers several advantages: *(i)* inaccuracies in water depth measurement become smaller in relative terms; *(ii)* the Reynolds number is higher, so that some artefacts (e.g. viscous effects) due to a change in the turbulence regime are minimized. Using a distorted model, it is even possible to keep both Reynolds and Froude numbers identical in prototype and in the model, with the same fluid (Finaud-Guyot et al., 2018). Distorted models have been used for a broad range of applications in fluvial and coastal hydraulics. Among others, Jung et al. (2012) applied scale factors $e_H = 120$ and $e_V = 50$ to study a floating island in a river; Wakhlu (1984) obtained comparable results on an undistorted



($e_H = e_V = 36$) and a distorted model ($e_H = 100$; $e_V = 17$) of a river division weir; However, since a distorted laboratory model corresponds to a representation of the prototype shrunk *differently* along the horizontal and the vertical directions, it may also lead to specific artefacts in the laboratory observations. For instance, Sharp and Khader (1984) highlighted distortion effects by comparing an undistorted ($e_H = e_V = 20$) and a distorted model ($e_H = 400$; $e_V = 100$) to study wave transmission and assess stone stability in a harbour.

### 2.2    Recent studies based on distorted models of urban flooding and potential artefacts

Figure 1 shows the horizontal and vertical scale factors used in recent laboratory studies of urban flooding at the district level. The grey shading in Figure 1a suggests conceptually that the larger the scale factors, the greater the expected scale effects; while Figure 1b indicates that using a strongly distorted model (i.e. a large ratio $e_H / e_V$) also leads to specific artefacts, which we refer to hereafter as *distortion effects*. In this regard, Chanson (2004) suggests to keep the ratio $e_H / e_V$ below 5 to 10 (orange dash-dot lines in Figure 1b).

An outdoor distorted model was used by Smith et al. (2016) to represent pluvial flooding in an urban district of relatively limited extent (Table 1). The scale factors were as low as $e_H = 30$ and $e_V = 9$ (distortion ratio: $e_H / e_V = 3.3$), thus minimizing potential scale effects. Similarly, Güney et al. (2014) used an outdoor distorted model of a large urban district (Table 1), with $e_H = 150$ and $e_V = 30$ (distortion ratio: $e_H / e_V = 5$), to represent dam-break-induced flood waves. The distortion ratio of these two models remains below the upper bound of 5 to 10 as recommended by Chanson (2004). Lipeme Kouyi et al. (2010), Araud (2012) and Finaud-Guyot et al. (2018) considered the same geometric configuration (Supplement 1), involving seven streets aligned along one direction, crossing seven other streets. The scale factors used by Lipeme Kouyi et al. (2010) were $e_H = 100$ and $e_V = 25$. The setup of Araud (2012) and Finaud-Guyot et al. (2018) contains substantial improvements compared to the initial setup of Lipeme Kouyi et al. (2010), mainly regarding the control of the inflow in each street separately; but it uses a considerably higher horizontal scale factor ($e_H = 200$) and, simultaneously, a smaller vertical scale factor ($e_V = 20$). This leads to a particularly high ratio $e_H / e_V$, equal to the upper limit of 10 suggested by Chanson (2004). If the model of Araud (2012) and Finaud-Guyot et al. (2018) was not distorted, the Reynolds numbers would have been about 30 times lower ($R \sim 1 \times 10^2 \div 1 \times 10^3$) than they actually are (Table 1).

### 2.3    Specific objective of the present study

While the motivations for using a large distortion ratio between the horizontal and vertical scale factors are not arguable (fit the model within a limited laboratory space, improve the accuracy of water depth measurement, maintain a sufficiently high Reynolds number), assuming no artefacts in experimental observations performed on a strongly distorted model may legitimately be questioned. Among other aspects, the complex three-dimensional flow structures observed in individual crossroads (Mignot et al., 2008, 2013; Rivière et al., 2011, 2014) suggest that "shrinking" the model vertically is likely to alter these flow structures, and hence also impair the representation of flow partition in-between the streets. The influence of strong distortion in laboratory scale models was investigated for some specific applications, such as in coastal engineering (Ranieri, 2007; Sharp and Khader, 1984), but it was never analyzed in the context of laboratory models of urban flooding.

Therefore, in this paper, we aim to evaluate the artefacts arising from the use of distorted laboratory models in experimental studies of urban flooding at the district level. We base our assessment on the reanalysis of two recent datasets, presented respectively by Araud (2012) and by Velickovic et al. (2017). The latter does not represent a realistic urban district but solely of regular grid of obstacles, to some extent similar to a network of streets. We focus on the influence of model distortion on the observed water depths and discharge partition in-between the streets.





## 3. Data and methods

### 3.1 Datasets

Two datasets were reanalyzed, corresponding respectively to an entire urban district and to a group of buildings. The former dataset was collected by Araud (2012) in the ICube laboratory in Strasbourg (France) and was also presented by

5 Arrault et al. (2016) and Finaud-Guyot et al. (2018) (Supplement 2: Figure S3a). The experimental model (5 m × 5 m) represents an idealized urban district of 1 km by 1 km at the prototype scale. It contains a total of 14 streets of various widths (0.05 m to 0.125 m) and 49 intersections (crossroads). The inflow discharge was controlled in each street individually and the outflow discharges were monitored downstream of each street. The uncertainties in the estimation of the outflow discharges are discussed in Section 4.1. For several steady inflow discharges, the water depths along the

10 centreline of streets were measured using an optical gauge (1 mm accuracy, Figures S4 to S7). Hereafter, we consider the experimental *runs* performed with a total inflow discharge $Q_m$ of 20, 60, 80 and 100 m³h⁻¹ in the laboratory model (Table 2). In each test, 50 % of the total inflow discharge was fed to the west face of the model and 50 % to the north face of the model. The specific inflow discharge was kept the same for each street of a given face, the outflow discharge was estimated from a calibrated rating curve (Araud, 2012). As detailed in Supplements 2 to 5, several measurements were

15 repeated, which allows appreciating the reproducibility of the tests.

The second dataset was collected by Velickovic et al. (2017) in the Hydraulic Laboratory of Université Catholique de Louvain, Belgium. A group of 5 × 5 square obstacles (buildings) of 0.30 m by 0.30 m was installed in a horizontal flume 36 m long and 3.6 m wide. Several layouts of obstacles were considered and we analyse here three of them (aligned with the channel) (Figure S3b), for which at least four to six experimental runs were conducted with various steady inflow

discharges. The three layouts differ by the distance in-between the obstacles (i.e. the street widths) in the direction normal to the main flow (0.0675 m, 0.10 m and 0.135 m). Profiles of water depth (accuracy: 0.1mm) were measured with movable ultrasonic probes along the centreline of the streets aligned with the main flow direction. In contrast, the flow partition in-between the streets was not measured by Velickovic et al. (2017).

### 3.2 Method

As sketched in Figure 2a, the initial interpretation of the various experimental runs of Araud (2012) and Velickovic et al. (2017) was that each model run corresponds to a different flooding scenario (i.e. a different total inflow discharge $Q_m$ into the urban district) represented with fixed scale factors $e_H$ and $e_V$. In contrast, in the present reanalysis of the laboratory dataset, we propose to consider that a single flooding scenario was represented in each laboratory model; but that the various experimental runs actually correspond to different vertical scale factors $e_V$. This new perspective is sketched in

Figure 2b.

As detailed hereafter, we followed a three-step procedure for the reanalysis, for each model:

1. select one experimental run, and assign to it plausible scale factors $e_H$ and $e_V$;
2. estimate the scale factor $e_V$ corresponding to each of the other experimental runs, assuming that $e_H$ remains unchanged and that all experimental runs simulate the same flood scenario;
3. upscale the experimental observations of each run to the prototype scale and compare them to each other.

### Step 1

We considered that Run 1 (Table 2) of Araud (2012) corresponds to a representation of a given flooding scenario in a strongly distorted scale model, consistently with the original values $e_H = 200$ and $e_V = 20$ reported by Arrault et al. (2016).





Similarly, for the dataset of Velickovic et al. (2017), we considered that the street width (0.10 m), in the model layout characterized by an intermediate street width, corresponds to 10 m in the prototype. This sets the horizontal scale factor to $e_H = 100$. Keeping $e_H = 100$ for the two other layouts leads to reproducing prototype street widths of 6.75 m for the narrow street layout and 13.5m for the wide street layout. As shown in Table 3, we also assumed that the experimentally

observed water depth (~ 0.3 m) for the highest inflow discharge $Q_m$ in the layout with an intermediate street width (Run 6) corresponds to about 4 m in the prototype (i.e. extreme flooding conditions such as induced by a dam break). This leads to a vertical scale factor $e_V = 13$ for this run (Table 3).

*Step 2*

For the dataset of Araud (2012), Runs 2 to 4 in Table 2 are now assumed to represent the same flooding scenario as Run1,

with the same horizontal scale factor $e_H$; but with adjusted vertical scale factors $e_V$. Similarly, all Runs in Table 3 (dataset of Velickovic et al., 2017) represent the same flooding scenario as Run 6 of the intermediate street layout; but with different vertical scale factors. For both datasets, the adjusted values of $e_V$ were derived as follows:

- The run selected in Step 1 was upscaled to the prototype scale, enabling the determination of the inflow discharge $Q_p$ in the prototype.

- Knowing the inflow discharge $Q_m$ for each of the other model runs, the adjusted vertical scale factor was calculated as: $e_V = (Q_p / Q_m / e_H)^{2/3}$, consistently with Froude similarity.

The values of $e_V$ derived from this procedure are detailed in Table 2 and Table 3 for the two datasets. The model distortion, expressed as the ratio $d$ between $e_H$ and $e_V$, varies between 3 and 7.7 in the dataset of Velickovic et al. (2017), and between 3.4 and 10 in the dataset of Araud (2012).

*Step 3*

Finally, for each model, the experimental observations of all model runs (measured water depths and outflow discharges) were upscaled to the prototype scale and compared. This approach is similar to the "scale series" method described by Heller (2011) and used by Erpicum et al. (2016) although here we vary only the vertical scale factor.

If there were no artefacts arising from the model distortion, the prototype scale predictions from the different experimental

runs should superimpose. In the following, we assess the magnitude of the distortion effects by comparing the upscaled observations from each experimental run with one selected reference, namely the experimental run corresponding to the weakest distortion (minimum value of $d$), for which we have the highest confidence in prototype event replication: Run 4 ($d = 3.4$) in the dataset of Araud (2012) and Runs 1 of each layout ($d = 3, 4.3, 4.9$) in the dataset of Velickovic et al. (2017).

**4.  Results**

We first present an estimation of the experimental uncertainties (Section 4.1); then, we detail the effect of model distortion on the upscaled water depths (Section 4.2). Finally, we present the effect of model distortion on the partition of outflow discharges (Section 4.3).

*4.1    Experimental uncertainties*

The measurement accuracy and the fluctuations of water surface are considered as the two main sources of uncertainty in the experimental records of water depths. For the dataset of Araud (2012), the accuracy of the optical gauge is about 1 mm. As shown in Supplement 2, the water depths were measured twice at some locations for Run 3 (60m³h⁻¹) and Run 2





($80 m^3 h^{-1}$). The repeatability of the tests is evaluated by comparing the repeated measurements. As detailed in Supplement 3, the difference in water depths between two measurements remains below 2 mm for 90 % of the dataset. Therefore, we estimate here the water depth measurement uncertainty at about 2 mm. Representative profiles of measured water depths (including repetitions) are displayed in Supplement 4 and they confirm the validity of this uncertainty estimate.

The discharge at the outlet of each street was estimated from a rating curve corresponding to a weir at the downstream end of dedicated measurement channels (located downstream of each street outlet). The water depth measurements were performed at least three times for each model run. As detailed in Supplement 5, comparing the repeated measurements demonstrates an excellent repeatability of the outflow discharge estimates (Figures S12 to S15). Hence, the main source of uncertainty in the outflow discharge estimates is assumed to stem from the accuracy of water depth measurement

(1 mm) over the measurement weirs (Araud, 2012). This leads to about 2 % to 6 % uncertainty in the outflow discharges, as shown in Figure S16.

For the second dataset, the accuracy of the probes, as described by Velickovic et al. (2017), is 0.1 mm. Velickovic et al. (2017) also reported the standard deviation of the recorded water depths at each probe location. We used this as a proxy for estimating the flow variability. Figure S9 in Supplement 3 shows the cumulative distribution function of this standard

deviation over all measured data in the three layouts and for each experimental run. For 90 % of the measurement points, the standard deviation may be as high as 2 to 6 mm, which reflects considerable fluctuations in the free surface, particularly for the higher flow rates.

### 4.2 Effect of model distortion on predicted water depths

*Dataset of Araud (2012)*

All measured water depths were upscaled to the prototype scale using the vertical scale factors defined in Table 2. As exemplified in Figure 3 for street 4 (and in Figures S17 and S18 in Supplement 6 for streets A, B, and C), the tests conducted with the strongest distortion ($d = 8.6$ and $d = 10$) lead to the lowest estimates of water depths at the prototype scale. Conversely, the highest predicted water depths correspond systematically to the upscaled measurements from the tests involving a weaker distortion ($d = 3.4$), except nearby the downstream boundary conditions where all predicted

water depths are close to each other. The tests conducted with $d = 7.1$ lead to intermediate estimates of water depths at the prototype scale. The reasons for these differences are discussed in section 5.1. Here, the uncertainty associated to the upscaled water depths is larger when the model distortion is limited since the absolute value of the measurement uncertainty remains the same in all model run; but a larger vertical scale factor is applied.

In Figure 4, Figure 5 and Table 4, we compare quantitatively the upscaled water depths obtained from the four different

model runs listed in Table 2. The model run with the weakest distortion is used as a reference for comparisons. The following observations can be made:

- The scatter plots in Figure 4 confirm that the water depths derived from the model with the lowest distortion are generally higher than those derived from the other model runs.

- The dash-dot purple lines (and corresponding labels) (Figures 4b, 4d, 4f) indicate the range containing 90 % of

35 the data. This shows that, despite the uncertainties in the measurements, the differences between the model runs increase very consistently as the model distortion increases. This consistent trend is also emphasized by the evolution of the mean difference (*MD* in Table 4) from – 0.058 to – 0.107 as $d$ varies from 7.1 to 10, as well as





by the cumulative distribution of the differences in upscaled water depths derived from the various model runs (Figure 5).

- The 5 % percentile of the differences between the models with varying distortions is of the order of – 20 to – 25 cm (Table 4). When compared to the order of magnitude of the absolute value of water depths (~ 2 m), the effect of model distortion in the tested configurations is of the order of 10 % of the upscaled water depths.

- The spatial distributions provided in Figure 4 also reveal that the influence of model distortion tends to increase in the upstream part of the urban district (i.e. closer to the north and west faces). Two reasons contribute to explain this: *(i)* the water depths in the downstream part are mainly controlled by the free outflow boundaries and remain therefore more similar whatever the distortion; and *(ii)* the cumulated effect of friction (expected to be underestimated by the more distorted models) is stronger in the upstream part of the flow.

*Dataset of Velickovic et al. (2017)*

Figure S19 in Supplement 7 displays the upscaled water depth profiles based on the dataset of Velickovic et al. (2017) and the scale factors defined in Table 3 for each of the three layouts. The upscaled water depths in the three model layouts differ substantially as the same discharge $Q_p$ is imposed ($h_p$ in narrow street: ~ 5 m; median street: ~ 3.5 m; wide street: ~ 2.5 m). Now, for each model layout separately, we compare the observations corresponding to various inflow discharges in the same way as for the dataset of Araud (2012). It is found that the upscaled water depths in the urban area become systematically lower as the model distortion increases. This is observed consistently for the three layouts of obstacles (narrow, intermediate and wide streets) and the differences arising from the change in model distortion greatly exceed the estimated experimental uncertainty. This result is also confirmed by the values of mean differences (MD) reported in Table 5, as well as by the cumulative distribution functions of the differences in water depths (Figure S20 in Supplement 7). In each layout, the model characterized by the lowest value of $d$ was taken as a reference for comparison.

*4.3   Effect of model distortion on outflow discharges*

For the dataset of Araud (2012), the upscaled values of the discharge at the outlet of each street is shown in Supplement 8 (Figure S21). Although the differences in the outflow discharge appear of the same order of magnitude as the experimental uncertainties, the variation of the outflow discharges with $d$ shows a consistent monotonous trend (increasing or decreasing) in almost all outlet streets. Since the mass balance remains unchanged at the level of the whole district, the direction of change in the outlet discharge varies from one street to the other. In Figure 6a, we compare the upscaled outflow discharge in each street to the corresponding value derived from the less distorted model ($d = 3.4$). The changes in the estimated outflow discharge when $d$ is varied are generally in the range of ± 10 % in most of streets (maximum ± 18 %).

The magnitude of model distortion effects may differ from one flow variable to the other (Heller, 2011). To be able to appreciate the relative influence of model distortion on the outflow discharge and on the water depths, we estimated for each street outlet (noted $i$) the change in upscaled water depth ($h_i / h_{ref}$) "associated" to the observed change in the outflow discharge ($Q_i / Q_{ref}$) when the model distortion is varied (from $d_{ref}$ to $d_i$). To do so, using Froude similarity leads to: $h_i / h_{ref} = ( Q_i / Q_{ref} )^{2/3} e_{V,i} / e_{V,ref}$. As shown in Figure 6, the results indicate that, in the present case, the magnitude of model distortion effects appear relatively comparable for the water depths and the outflow discharges (of the order of ± 10 %).





## 5. Discussion

### 5.1 Relative importance of the main causes of distortion effects

The differences in the upscaled water depths as a function of the model distortion $d$ may result from differences in localized flow features (e.g. local head losses at street intersections) as well as from more distributed effects, which in the present case are likely to dominate only along the mainly one-dimensional flow regions (typically within a street). The latter effects may be categorized into three types:

- differences in the relative importance of the bottom roughness,
- differences in the relative importance of viscosity effects,
- differences in the aspect ratio of the flow section, affecting notably the secondary currents and velocity profiles of streamlines velocity.

The third type of effects is probably dominant, specifically at street intersections, mainly as a result of the change in the aspect ratio of the flow section. However, the available datasets, involving only observed water depths and discharges at the model outlets, do not enable investigating in detail these more localized effects. This could be achieved by means of new experiments with detailed velocity measurements within the urban district. 2D and 3D computational modelling may also be useful in this respect, the former enabling to account for the spatial variations of flow characteristics and the latter giving full access to the complex flow fields developing at the street intersections.

Nonetheless, we appreciate here the influence of the first two types of effects, of relevance in mainly one-dimensional flow regions. To do so, we estimate the energy slope $S_{f,p}$ at the prototype scale in the various model configurations based on Darcy-Weisbach equation:

$$S_{f,p} = \frac{e_V}{e_H} S_{f,m} = \frac{e_V}{e_H} \frac{f_m}{8} \frac{V_m^2}{gR_{H,m}} = \frac{1}{8} f_m \left( \mathsf{R}_m, \frac{k_{s,m}}{R_{H,m}} \right) \mathsf{F}^2 \frac{h_m}{R_{H,m}} \frac{e_V}{e_H}$$

Effect of viscosity ⎯⎯⎯⎯

Effect of relative roughness height

Effect of aspect ratio

(2)

in which the friction coefficient $f_m$ can be computed as a function of the Reynolds number $\mathsf{R}_m$ and the relative roughness height $k_{s,m} / R_{H,m}$ using the explicit approximation of Colebrook-White formula given by Yen (2002). The roughness height $k_{s,m}$ was taken equal to $10^{-5}$ m to represent the smooth bottom and walls of the experimental setups.

Figure S22 shows the values of parameters $\mathsf{R}_m$, $k_{s,m} / R_{H,m}$, $f_m$, $h_m / R_{H,m}$, and $\mathsf{F}$ representative of the flow conditions at the street inlets in the various experimental runs of Araud (2012). Given the values of $\mathsf{R}_m$ and $k_{s,m} / R_{H,m}$, the flow is in transitional regime in the Moody diagram. All these parameters change substantially when the model distortion $d$ is varied (Figure S23). As $d$ is increased, the friction coefficient $f_m$ decreases systematically ($f_{i,m} / f_{ref,m} \sim 0.8$), due to the joint effect of a higher Reynolds number and a lower relative roughness height ($k_{s,m} / R_{H,m}$). Similarly, the Froude number tends to slightly increase with the model distortion. In contrast, parameter $h_m / R_{H,m}$ shows a considerable systematic increase (1.6 ~ 2.3) as $d$ increases, due to the change in the aspect ratio of the flow section. The resulting energy slope $S_{f,m}$ in the inlet streets in each model run and the corresponding upscaled values $S_{f,p}$ are presented in Figure 7. The energy slope is lower in major streets (4, C, F) and in the straight streets (A, B). The energy slope $S_{f,m}$ in the model increases monotonously with the model distortion (mainly due to the increased wetted perimeter), whereas the energy slope $S_{f,p}$ at the prototype scale declines as $d$ is increased (Figure 7b). This results from a "competition" between the increase in $S_{f,m}$ and a decrease in the ratio $e_V/e_H$, as $d$ increases. This latter effect appears dominant in the mainly one-dimensional flow regions.





*5.2    Reference used for comparison*

Here, we used the "less distorted" models as a reference for our comparisons since the aim is to assess the influence of model distortion. However, given that we simply reanalysed already existing datasets obtained in experiments which were not designed in the first place for the sake of investigating the effect of model distortion, these "less distorted"

models are also those characterized by the largest vertical scale factors, hence enhancing other artefacts such as stronger viscosity effects (see markers "**\***" in Figure 1). Also, the upscaled measurement uncertainties are maximum in this case, as can be seen in Figure 3. Therefore, the present study needs to be complemented by new tailored experiments involving a model series (Heller 2011) with various levels of distortion and including a valid reference model, i.e. at least one model without distortion and characterized by sufficiently-small scale factors so that viscosity effects and experimental

uncertainties are kept minimum. This may be achieved by considering a model series in which the vertical scale factor $e_V$ is kept constant in all models, while only the horizontal scale factor $e_H$ is varied. This approach contrasts with the procedure followed here, in which $e_H$ was constant and $e_V$ was varied to take benefit of existing datasets.

**6.    Conclusion**

Laboratory scale models representing flooding of an entire urban district are usually distorted, in the sense that the vertical

scale factor ($H_p$ / $H_m$) is often considerably smaller than the horizontal one. This paper evaluates the influence of the model distortion on the upscaled values of water depths and outflow discharge for relatively extreme flood conditions (water depth and flow velocity of the order of 1~2 m and 0.5 ~ 2 m/s at the prototype scale). Two existing experimental datasets were reanalysed for this purpose. The results show that the stronger the model distortion, the lower the values of the upscaled water depths, whereas the influence on flow partition at the outlet of the street appears more complex.

The change in the upscaled water depths was found of the order of 10 % when the distortion of the model is varied by a factor 3. Moreover, about 10 % deviation was also obtained for the outflow discharges. This is of the same order as the influence of small-scale obstacles, as reported by Bazin et al. (2017).

For the water depths, the effect of varying the model distortion was found generally larger than the measurement uncertainties, whereas both effects are comparable for the discharge at the street outlets in the tested configurations. Note

that the relative measurement uncertainties are smaller in distorted models than in undistorted models.

The uncertainty in the upscaled water depths is considerably larger in the models with limited distortion, because they also correspond to the highest vertical scale factors. Using those results as a reference seems sensible to assess the effect of distortion since they correspond to the lowest values of *d*; but at the same time it is somehow problematic to use as a reference the results with the highest uncertainty (e.g., Figures S17 to S18). To overcome this issue, future experimental

research should involve "model series" in which the horizontal scale factor $e_H$ is reduced without changing the vertical scale factor $e_V$. This will enable reducing the influence of distortion without increasing the error in the estimated water depths; but this requires the collection of new experimental measurements, which are presently lacking. More controlling parameters should also be considered, such as the bottom slope, and computational modelling should complement the laboratory experiments.

**7.    Acknowledgements**

This research was partly funded through a grant for Concerted Research Actions (Grant n°ARC 13-17/01) financed by the Wallonia-Brussels Federation. The Authors gratefully acknowledge Prof. Sandra Soares-Frazão(Université Catholique de Louvain, Belgium) for sharing the experimental datasets.





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

**Tables**

Table 1 Recent laboratory experiments of urban flooding at the district level.

| References | Prototype spatial extent (km) | Model spatial extent (m) | Horizontal scale factor $e_H$ | Vertical scale factor $e_V$ | Reynolds number R in the model |
|---|---|---|---|---|---|
| LaRocque et al. (2013) | $0.28 \times 0.124$ | $5.6 \times 2.48$ | 50 | 50 | $\sim 7 \times 10^4$ |
| Testa et al. (2007) | $0.16 \times 0.12$ | $1.6 \times 1.2$ | 100 | 100 | $2 \div 4 \times 10^5$ |
| Ishigaki et al. (2003) | $2 \times 1$ | $20 \times 10$ | 100 | 100 | $\sim 7 \times 10^3$ |
| Smith et al. (2016) | $0.375 \times 0.15$ | $12.5 \times 5$ | 30 | 9 | $7 \div 9 \times 10^4$ |
| Güney et al. (2014) | $\sim 2.2 \times 2.2$ | $\sim 15 \times 15$ | 150 | 30 | $2 \div 10 \times 10^5$ |
| Lipeme Kouyi (2010) | $1 \times 1$ | $10 \times 10$ | 100 | 25 | $0.5 \div 5 \times 10^4$ |
| Araud (2012) and Finaud-Guyot et al. (2018) | $1 \times 1$ | $5 \times 5$ | 200 | 20 | $0.8 \div 4 \times 10^4$ |

15    Table 2 Initial interpretation of the laboratory model runs as various flooding scenarios represented with fixed scale factors (Araud, 2012; Finaud-Guyot et al., 2018) vs. interpretation in the present reanalysis, involving a single flood scenario represented with various vertical scale factors. Notations: $Q_m$, $Q_p$ refer to the total inflow in the laboratory model and in the prototype urban district, respectively.

| Runs | $Q_m$ (m³h⁻¹) | Initial interpretation by Araud (2012): fixed scale factors | | | Interpretation in the present reanalysis: fixed flood scenario; but varying vertical scale factor | | | |
|---|---|---|---|---|---|---|---|---|
| | | $Q_p$ (m³s⁻¹) | $e_H$ (-) | $e_V$ (-) | $Q_p$ (m³s⁻¹) | $e_H$ (-) | $e_V$ (-) | $d = e_H / e_V$ (-) |
| Run 1 | 100 | 497 | | | | | 20 | 10 |
| Run 2 | 80 | 398 | 200 | 20 | 497 | 200 | 23 | 8.6 |
| Run 3 | 60 | 298 | | | | | 28 | 7.1 |
| Run 4 | 20 | 99 | | | | | 58 | 3.4 |





Table 3 Interpretation of the dataset of Velickovic et al. (2017) in the present reanalysis. Notations $Q_m$ and $Q_p$ refer to the total inflow discharge in the laboratory model and in the prototype, respectively.

| Layouts | Runs | $Q_m$ (ls$^{-1}$) | $Q_p$ (m$^3$s$^{-1}$) | $e_H$ (-) | $e_V$ (-) | $d = e_H / e_V$ (-) |
|---|---|---|---|---|---|---|
| Narrow streets along the main flow direction (width: 0.0675 m) | Run 1 | 25 | | | 33.4 | 3 |
| | Run 2 | 35 | 483 | 100 | 26.7 | 3.7 |
| | Run 3 | 41 | | | 24.0 | 4.2 |
| | Run 4 | 50 | | | 21.0 | 4.8 |
| Intermediate streets width along the main flow direction (width: 0.10 m) | Run 1 | 43 | | | 23.3 | 4.3 |
| | Run 2 | 58 | | | 19.1 | 5.2 |
| | Run 3 | 63 | 483 | 100 | 18.0 | 5.5 |
| | Run 4 | 75 | | | 16.1 | 6.2 |
| | Run 5 | 86 | | | 14.7 | 6.8 |
| | Run 6 | 103 | | | 13.0 | 7.7 |
| Wide streets along the main flow direction (width: 0.135 m) | Run 1 | 52 | | | 20.5 | 4.9 |
| | Run 2 | 64 | | | 17.9 | 5.6 |
| | Run 3 | 75 | 483 | 100 | 16.1 | 6.2 |
| | Run 4 | 80 | | | 15.4 | 6.5 |
| | Run 5 | 92 | | | 14.0 | 7.1 |
| | Run 6 | 99 | | | 13.3 | 7.5 |

Table 4 Differences between upscaled water depths derived from various runs of Araud (2012).

| Indicators | Run 3 ($d$ = 7.1) vs. Run 4 ($d$ = 3.4) | Run 2 ($d$ = 8.6) vs. Run 4 ($d$ = 3.4) | Run 1 ($d$ = 10) vs. Run 4 ($d$ = 3.4) |
|---|---|---|---|
| $MD$* (m) | -0.058 | -0.088 | -0.107 |
| 95 % percentile (m) | 0.077 | 0.075 | 0.060 |
| 5 % percentile (m) | -0.181 | -0.237 | -0.247 |

5    * $MD$ stands for Mean Difference



Table 5 Differences between upscaled water depths derived from various runs of Velickovic et al. (2017).

| Layouts | Runs | *MD* (m) | 95 % percentile (m) | 5 % percentile (m) |
|---|---|---|---|---|
| Narrow street | $d = 3.7$ vs. $d=3$ | -0.27 | -0.17 | -0.36 |
| | $d = 4.2$ vs. $d=3$ | -0.32 | -0.22 | -0.45 |
| | $d = 4.8$ vs. $d=3$ | -0.52 | -0.50 | -0.66 |
| Median street | $d = 5.2$ vs. $d=4.3$ | -0.24 | -0.07 | -0.40 |
| | $d = 5.5$ vs. $d=4.3$ | -0.37 | -0.21 | -0.71 |
| | $d = 6.2$ vs. $d=4.3$ | -0.47 | -0.24 | -0.62 |
| | $d = 6.8$ vs. $d=4.3$ | -0.46 | -0.38 | -0.83 |
| | $d = 7.7$ vs. $d=4.3$ | -0.77 | -0.43 | -1.19 |
| Wide street | $d = 5.6$ vs. $d=4.9$ | -0.45 | -0.30 | -0.79 |
| | $d = 6.2$ vs. $d=4.9$ | -0.59 | -0.44 | -0.92 |
| | $d = 6.5$ vs. $d=4.9$ | -0.63 | -0.52 | -0.97 |
| | $d = 7.1$ vs. $d=4.9$ | -0.72 | -0.55 | -1.25 |
| | $d = 7.5$ vs. $d=4.9$ | -0.75 | -0.57 | -1.25 |

* *MD* stands for Mean Difference

**Figures**

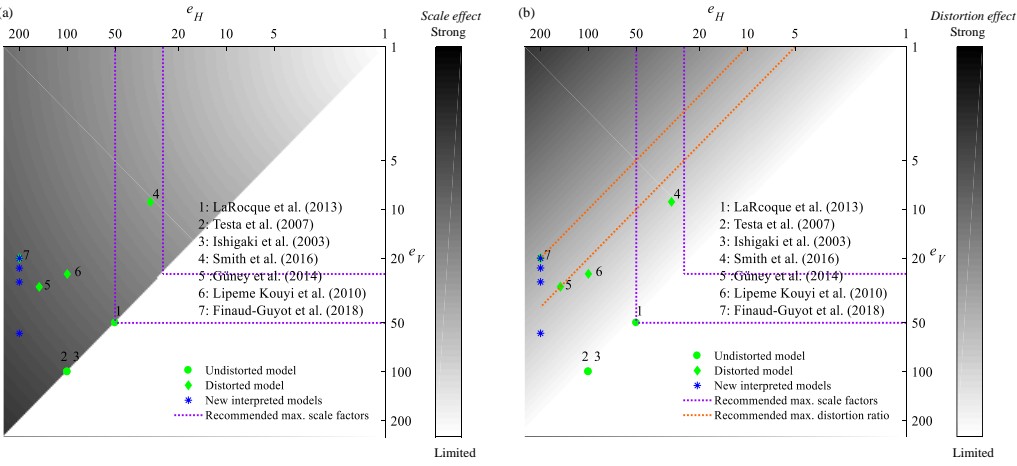

Figure 1: Recent laboratory models of urban flooding at the district level as a function of the horizontal and vertical scale factors, $e_H$ and $e_V$. The grey shading reveals qualitatively the possible magnitude of (a) scale effects and (b) distortion effects; a range of maximum scale factors (purple lines) and distortion ratio (orange lines) were recommended by Chanson (2004).





(a) Initial interpretation by Araud (2012) and Velickovic et al (2017):
Single value of the scale factors $e_H$ and $e_V$; but various flooding scenarios

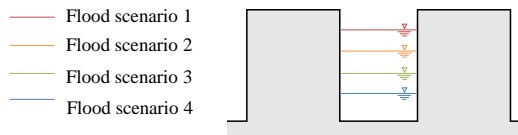

(b) Interpretation in the present reanalysis:
Single flooding scenarios; but various values of the vertical scale factor $e_V$

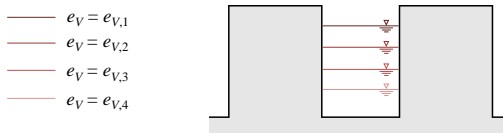

Figure 2: (a) Sketch of the initial interpretation of the laboratory model runs as various flooding scenarios represented with fixed scale factors. (b) Sketch of the interpretation in the present reanalysis, involving a single flood scenario represented with various vertical scale factors $e_V$.

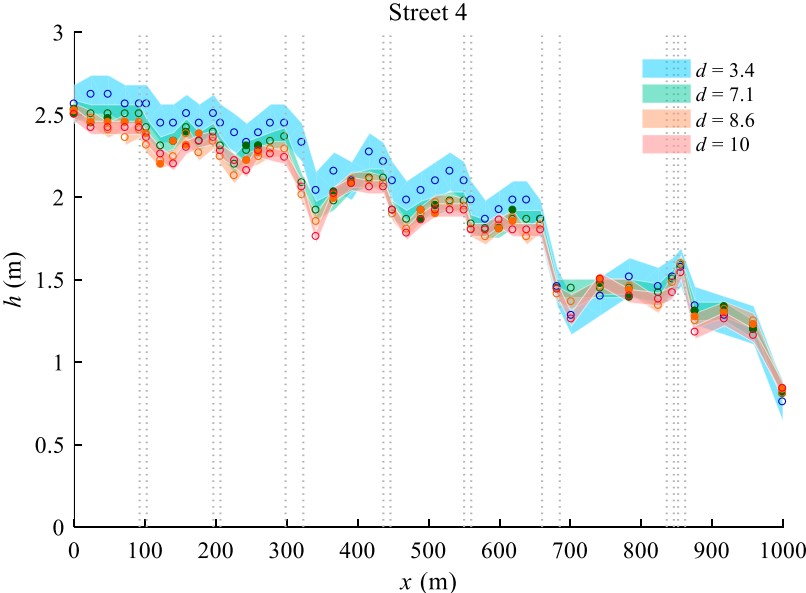

Figure 3: Upscaled water depth profiles in prototype in street 4 of Araud (2012) predictions, colour shade represents the upscaled measurement uncertainty.

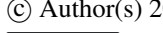


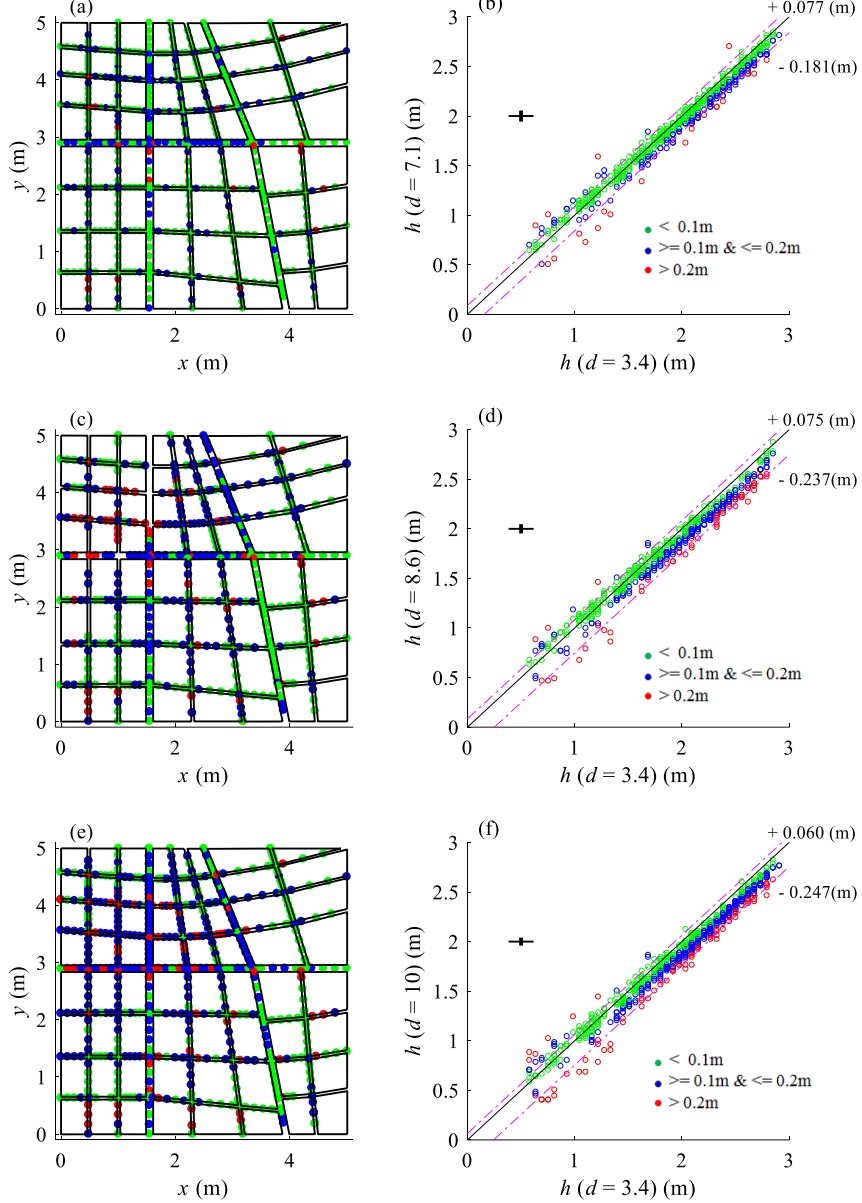

Figure 4: Spatial distributions (a, c, e) and scatter plots (b, d, f) of the differences between the upscaled water depths in Runs 3 ($d$ = 7.1), 2 ($d$ = 8.6) and 1 ($d$ = 10) and those derived from Run 4 ($d$ = 3.4) from Araud (2012). The dash-dot purple lines indicate the range containing 90 % of the data.




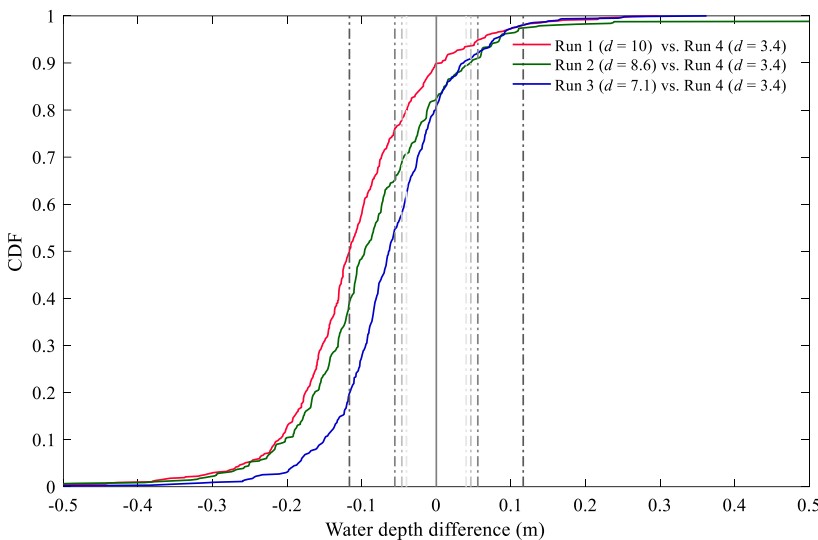

Figure 5: Cumulative distribution function of the differences between upscaled water depths from the various model runs of Araud (2012), the grey lines represent the upscaled measurement uncertainties with various $e_V$ (the more $e_V$ larger, the more line colour deeper).

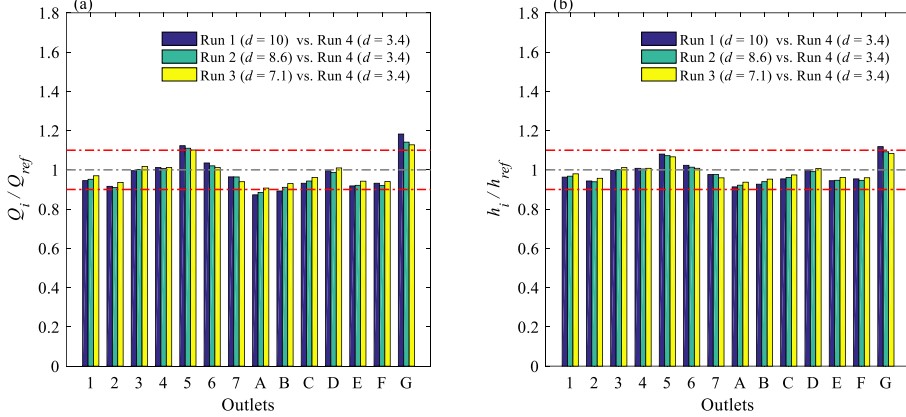

Figure 6: (a) Upscaled outflow discharge in each street compared to the corresponding value deduced from the observations in the less





distorted model (d = 3.4) and (b) ratio of the associated water depths from Araud (2012) and Finaud-Guyot et al. (2018). Red dash dot lines represent the ± 10 % range.

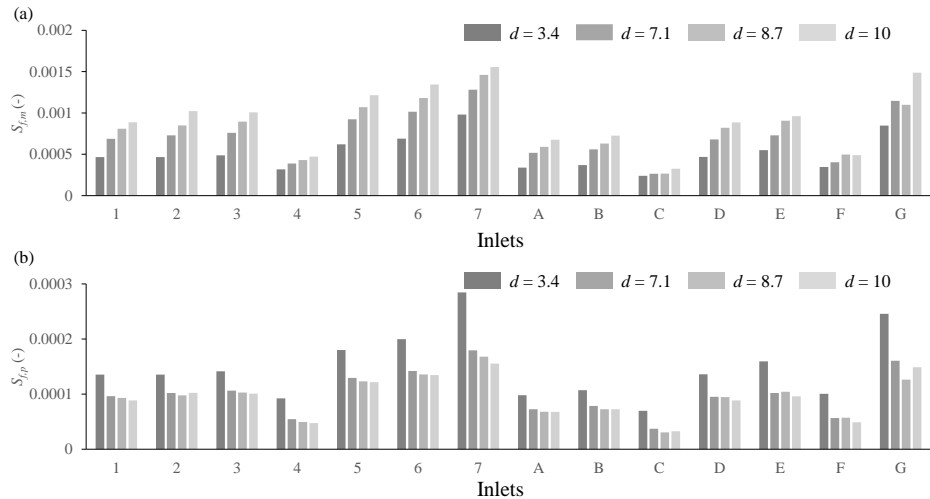

Figure 7: Distortion effect on energy slope of (a) scale models and (b) the model in prototype (dataset of Araud (2012)).

