# Peer review of "Laboratory modelling of urban flooding: strengths and challenges of distorted scale models"

_Hydrology and Earth System Sciences, 2018_

## Referee Comment (RC1) · Anonymous Referee #1 · 12 Nov 2018

General Comments The authors present an interesting paper on scale effects in the context of the modelling of urban flooding using geometrically distorted models, based on the reanalysis of existing lab data. I am not aware of any similar analysis done in this context, and the paper will be of significant interest to those undertaking such studies. The paper is well written and clear and I would certainly recommend publication. I have made some suggestions for potential improvements below. Specific Comments 1. The abstract (L17) contains a comparison of uncertainties from scale effects to those from hydrological data. This analysis does not seem to be in the paper or conclusions. The abstract should not contain information that is not in the full paper; hence this should be removed or added to the paper. Please also note the typo on the last line. 2. Introduction (2nd paragraph) – The authors point out the difficulty in obtaining velocity data in

field conditions with which to validate flood models. However point velocity data is also uncommon for lab studies of urban flooding (for example due to the low flow depths and the difficulty in installing equipment such as ADV probes). Indeed the studies the authors use is based on flow and depth data. Some studies which provide velocity data in a lab setting have recently come available (e.g. Martins et al 2018, WRR) and the authors may at some point in the paper wish to consider the benefits of velocity + depth data for model validation or scale model studies, as well as clarifying the relative rarity of velocity data in lab or field studies. 3. P2 L16 – It would benefit the introduction if the authors could be a bit more specific regarding what they mean by 'artefacts'. Perhaps with a specific example of how these might arise in scaled experiments, what is the physical cause etc. 4. P2 L 25 – Chanson gives some specific recommendations, however it would be good if the basis for these recommendations was briefly mentioned. 5. P3-P4 The authors provide a good review of physical models and scaling issues in the simulation of surface urban flood flows. The authors should briefly note that studies of 'dual drainage' (surface and sewer flows) also exists however the scaling issues are somewhat complicated by the combination of pressurised pipe and surface flow. This generally posed some additional limitations on these studies (either associated with the hydraulic conditions which can be simulated, or the minimum size of the model). 6. P5 – For both data sets the authors do not discuss the quantification of the inflow discharge. It seems to be assumed that the error within the quantification of this discharge is negligible. Some justification for this assumption should be given. 7. The treatment of frictional resistance within scaled models is an interesting topic which the authors mention only briefly within the discussion. Given such models are generally constructed from smooth materials can the authors comment further on the representation of roughness within scaled models and any implications? Why is friction expected to be underestimated in more distorted models? 8. P9 L11 – Can the authors comment on why they expect the third effect to be probably dominant? Perhaps this should be phased 'The third of these effects….'

Technical comments 1. P2 L2 'then ever observed in the past' should be removed (too

many points in one sentence) 2. P3 L6 ' replace 'were' with 'have been' 3. P3 L11 – Seems to have a divide symbol? 4. P4 L34 – Replace 'it was never' with 'has not to date been'

---

## Referee Comment (RC2) · Anonymous Referee #2 · 19 Nov 2018

GENERAL COMMENTS The manuscript presents an interesting study about the influence of horizontal/ vertical distortions of lab hydraulic models – to the best of my knowledge this is a novel investigation in the topic. The study is based on two existing datasets that were generated with other objectives in mind and, for this reason, the results are (somewhat) not fully conclusive, as the authors mention at the later sections of the manuscript. Nevertheless, in my opinion, the results are relevant to improve urban drainage lab modelling, and perhaps more important, the study identifies the need of new lab experiments to improve the understanding of urban drainage systems hydraulics. The manuscript is well written and clear. My main concern is the lack of consideration of uncertainties in the hydraulic measurements and their influence in the conclusions of the study. Below I suggest a few points that the authors may consider

[Figure]

to improve the quality of the manuscript

SPECIFIC COMMENTS Abstract: the meaning of "artefacts" should be presented, otherwise it is too vague. P1,L34: the authors refer to aerial images as a source for reference data. Although this is in theory valid, in many urban flooding situations the events are of short duration making capture of aerial data (e.g. from satellite and un- or manned aircraft sensors) challenging. P2,L6: the authors may also want to refer other recent lab experiments that were conducted at real scale (e.g. Moy de Vitry et al., 2017 – ESSD). These, of course, do have other challenges than those investigated in the present manuscript, but should also be discussed in the context of urban drainage hydraulic lab experiments. P2,L11: the authors refer to "both scales" at this point of the manuscript without yet having defined it. I assume the authors are referring to horizontal and vertical scales. Please clarify. P3,L35: "Using a distorted model...". It seems that the sentence should start with something like "By using a distorted...". P5,L19: "... at least four to six...". I believe the authors can be more specific here. The mentioned experiment was conducted in 2017, so it should not be difficult to know exactly how many runs were carried out! P5,L22: "In contrast, ...". I cannot understand to what (sentence or idea) this sentence is contrasting to. Please consider rephrasing it. P7,L16: "... may be..."the use of this words does not sound very scientific. Please consider a different construction, e.g. "is between" P8,L12-17: this seems to belong to the methodology section. It is definitely not a result. Please consider moving these sentences to another section. P8,L26: "... almost...". This is not a very scientific word. How many? What %? P8,L29: "... most...". This is not a very scientific word. How many? What %? Figure 6a: does the figure report V (V=SUM Q) or Q? this should be made clear. Also, if it is Q, the variation can be included (error bars?). P9,L9: "Aspect ratio". Should this be defined? Is it similar to hydraulic radius? Please explicitly define the meaning of "aspect ratio".
* * *

---

## Author Comment (AC1) · 20 Nov 2018

Xuefang Li[1], Sébastien Erpicim[1], Martin Bruwier[1], Emmanuel Mignot[2], Pascal Finaud-Guyot[3], Pierre Archambeau[1], Michel Pirotton[1], Benjamin Dewals[1]

[1] Hydraulics in Environmental and Civil Engineering (HECE), University of Liège (ULiège), Liège, 4000, Belgium
[2] LMFA, CNRS-Université de Lyon, INSA de Lyon, Lyon, 69100, France
[3] ICube laboratory (UMR 7357), Fluid mechanics team, ENGEES, Strasbourg, France

*Correspondence to*: Xuefang Li (Xuefang.li@uliege.be)

We would like to thank Referee 1 for his/her positive feedback and useful comments on how to improve the manuscript. We provide hereafter a point-by-point response to the Referee's comments.

1. As to the first comment about the comparison of uncertainties from distortion effects to those from hydrological data (L17 in the Abstract), we agree with the Referee that this should not be included there since it is not supported by factual material in the manuscript. Hence, L17 of the Abstract will be deleted in the revised manuscript as suggested by the Referee.

2. A second point raised by the Referee relates to the difficulty of conducting point velocity measurements on experimental models of urban flooding at the district level, with which we concur. Indeed, most existing lab studies representing urban flooding at the district level include only depth and discharge measurements (Finaud-Guyot et al., 2018) or additionally *surface* flow measurements (LaRocque et al., 2013). Only few studies report point velocity measurements for urban flooding at the district level (Güney et al., 2014; Park et al., 2013; Smith et al., 2016), while Zhou et al. (2016) used PIV to provide insights into the flow field in the wake of an array of buildings. As pointed out by the Referee, more studies involving detailed velocity measurements are coming up (e.g. Martins et al. (2018)), particularly for more local analyses (e.g. at the level of a single manhole, not at the level of an entire district). This issue will be discussed explicitly in the revised manuscript. We suggest to emphasize the need for pointwise velocity measurements as a perspective in the Conclusion of the revised manuscript.

3. As suggested by the Referee in his/her Comment 3, we will be more specific in the revised manuscript with respect to what we call "artefacts" in scale models. They correspond to deviations between up-scaled model measurements and real-world prototype observations due to governing non-dimensional parameters (i.e. force ratios) which are not identical between the model and its prototype (Heller, 2011). This may include alteration of the flow regime (transition vs. complete turbulent), or of the relative importance of frictional resistance …

4. Although the recommendations by Chanson (2004) look very reasonable, little background information is provided by Chanson (2004) to support the stated recommendations (Reynolds number above 5,000 in the model, distortion ratio below 5 to 10). We will nonetheless investigate this point and report our findings in the revised version of the manuscript.

5. In his/her Comment 5, the Referee highlights that scale effects in "dual drainage" models become even more intricate due to the combination of pressurized pipe flow and surface flow. Since the body of the present manuscript focuses on surface flow in a network of streets (not dual drainage), we propose to include a brief discussion on dual drainage as a perspective in the Conclusion of the revised manuscript. We already have a sentence stating that "More controlling parameters should also be considered, such as the bottom slope …". We will add a short paragraph on the importance of dual drainage for urban flooding and the associated experimental modelling challenges.

6. The Referee is right that information on the inflow discharge uncertainties were missing in the manuscript. For the dataset of Araud (2012), the uncertainty on inflow discharge is estimated at about 1 % (Finaud-Guyot et al., 2018), while in the study of Velickovic et al. (2017), the accuracy of the flowmeter measuring inflow discharge is of 1 ls$^{-1}$.

7. As stressed in the Referee's Comment 7, the representation of frictional resistance in scale models of urban flooding is particularly challenging. In this respect, we will clarify the following aspects.

   a. When we write that "friction is expected to be underestimated in more distorted models", we mean "underestimated" compared to the case of an *undistorted model* with similar bed material (not necessarily compared to the prototype!). This is true because the more distorted the model, the higher the Reynolds number and the lower the relative roughness ($k_{s,m} / R_{H,m}$). Both of these effects contribute to lower the friction factor, as confirmed by the experimental results in Figure S23.

   b. In the original manuscript, we hardly explain that, even without distortion, frictional resistance may not be properly represented in the scale model, particularly when smooth material is used to construct the bottom and the walls of the model. There are two competing effects (lower Reynolds number in the model compared to the prototype; but also lower relative roughness) which, in general, hampers a definite prediction on whether frictional resistance is over- or under-estimated compared to the prototype. We will discuss this explicitly in the revised manuscript.

   We will then highlight in the revised manuscript that, in distorted models, frictional resistance becomes indeed relatively smaller compared to the case of an undistorted model; but not necessarily compared to the prototype scale.

8. In his/her Comment 8, the Referee is right that "the third type of effects" being dominant is not well supported at this stage. This would require additional detailed measurements and/or interpretation based on computational modelling. Therefore, this paragraph will be rephrased in the revised manuscript.

All Technical comments by the Referee will be accounted for in the revised version of the manuscript.

On behalf of all authors,

Xuefang LI

**References**

5    In the reference list below, we include only the references which were not cited in the original manuscript.

Martins, R., Rubinato, M., Kesserwani, G., Leandro, J., Djordjevic, S. and Shucksmith, J. D.: On the Characteristics of Velocities Fields in the Vicinity of Manhole Inlet Grates During Flood Events, Water Resources Research, 54(9), 6408–6422, doi:10.1029/2018WR022782, 2018.

Park, H., Cox, D. T., Lynett, P. J., Wiebe, D. M. and Shin, S.: Tsunami inundation modeling in constructed environments: A
10   physical and numerical comparison of free-surface elevation, velocity, and momentum flux, Coastal Engineering, 79, 9–21, doi:https://doi.org/10.1016/j.coastaleng.2013.04.002, 2013.

Zhou, Q., Yu, W., Chen, A. S., Jiang, C. and Fu, G.: Experimental Assessment of Building Blockage Effects in a Simplified Urban District, in Procedia Engineering, vol. 154, pp. 844–852., 2016.

---

## Author Comment (AC2) · 26 Nov 2018

Xuefang Li[1], Sébastien Erpicim[1], Martin Bruwier[1], Emmanuel Mignot[2], Pascal Finaud-Guyot[3], Pierre Archambeau[1], Michel Pirotton[1], Benjamin Dewals[1]

[1] Hydraulics in Environmental and Civil Engineering (HECE), University of Liège (ULiège), Liège, 4000, Belgium
[2] LMFA, CNRS-Université de Lyon, INSA de Lyon, Lyon, 69100, France
[3] ICube laboratory (UMR 7357), Fluid mechanics team, ENGEES, Strasbourg, France

*Correspondence to*: Xuefang Li (Xuefang.li@uliege.be)

We gratefully acknowledge Referee 2 for his/her supportive comments and insightful suggestions to improve the manuscript. We provide hereafter a point-by-point response to the Referee's comments.

**General comments**

We concur with Referee 2 regarding the importance of considering uncertainties in hydraulic measurements. However, we believe that these uncertainties are already carefully reported and discussed in the manuscript. Particularly, Section 4.1 is entirely devoted to the evaluation of the experimental uncertainties, with additional details provided in Supplements 2, 3, 4 and 5, as briefly summarized hereafter.

- To estimate the uncertainties affecting the experimentally measured *water depths*, we account both for the instrument accuracy and for the repeatability of the tests (flow variability). The latter is detailed in Supplements 3 and 4. The water depth profiles provided in Supplement 4 also display the corresponding uncertainties (~ 2 mm). Importantly, Figs. 3 and 5 compare the influence of the experimental uncertainties and the effect of model distortion on the *upscaled* water depths. See also L26-28 and L35-36 on p. 7: "*... despite the uncertainties in the measurements, the differences between the model runs increase very consistently as the model distortion increases ...*", as well as L18-19 on p. 8: " *… the differences arising from the change in model distortion greatly exceed the estimated experimental uncertainty*."

- The uncertainties on estimated outflow discharges are also reported in Section 4.1 and details on the repeatability of the measurements are presented in Supplement 5 (see also L7-8 on p. 7: "*comparing the repeated measurements demonstrates an excellent repeatability of the outflow discharge estimates*"). The standard deviation of estimated outflow discharges is displayed in Fig. S21, which supports the comments formulated in Section 4.3.

In addition, all our main results are presented with the corresponding uncertainties (Fig. 3, Supplements 6, 7 and 8). Nonetheless, we understand from the Referee's comment that this consideration for the uncertainties should be made more explicit. We will update the revised manuscript accordingly.

**Specific comments**

1. As already mentioned in our response to Referee 1, we will specify what we mean by "artefacts". They correspond to deviations between up-scaled model measurements and real-world prototype observations due to governing non-dimensional parameters (i.e. force ratios) which are not identical between the model and the prototype (Heller, 2011). This may include alteration of the flow regime (transition vs. complete turbulent), or of the relative importance of frictional resistance …

2. We agree with the referee about the challenges associated to the use of aerial imagery to document (short duration) urban flooding. We will mention it explicitly in the revised manuscript. We will also highlight the potential of CCTV cameras to deliver valuable information on the dynamics of urban flooding.

3. In Section 2 (Background and motivation) of the revised manuscript, we will refer to the study by Moy de Vitry et al. (2017), which explores the potential of unconventional data (video data and computer vision) for monitoring urban flooding.

4. P2, L11: The wording "both scales" does not refer to the horizontal and vertical scales; but it refers to the "prototype" and the "model". This will be clarified in the revised manuscript.

5. P5, L19: As suggested by the Referee, we know indeed the exact number of experimental runs. They are all listed in Table 3. This will be made clear in the revised manuscript.

6. On P5, L22, we mean "In contrast with the dataset of Araud (2012) …", the dataset of Velickovic et al. (2017) does not include discharge data. We will rephrase this sentence.

7. P8, L12-17: These sentences do present results, because *upscaled* water depths (not the original experimental observations) are obtained by applying the methodology developed in the present study.

8. P8, L26 and P8, L29: The words "almost" and "most" used in the Section 4.3 refer to a general trend; but we agree with the Referee that it deserves being quantified. Hence, a detailed quantification of the distortion effect on the outflow discharges will be included in the revised manuscript.

9. Figure 6a presents the outflow discharge in each street, as described in Section 4.3. The variations of outflow discharge are presented in Fig. S21 in the form of error bar, as recommended by the Referee.

10. The *aspect ratio* is defined as the ratio between the water depth and the street width. This definition is indeed missing in the manuscript and it will be added in the revised version.

All other Technical comments (language, rewording …) will be accounted for in the revised version of the manuscript.

On behalf of all authors, Xuefang LI

**References**

We include below only a reference which was not cited in the original manuscript.

Vitry, M. Moy de, Dicht, S. and Leitao, J. P.: floodX: urban flash flood experiments monitored with conventional and alternative sensors, Earth System Science Data, 9(2), 657–666, doi:10.5194/essd-9-657-2017, 2017.

5

---

## Author Response (AR1)

**COVER LETTER**

Dear Editor,

Thank you very much for handling our manuscript: "Laboratory modelling of urban flooding: strengths and challenges of distorted scale models". We deeply appreciate the Referees comments and suggestions, which have enabled us to further improve the quality of the manuscript.

In particular, we have now

- defined what we mean by "artefacts";
- highlighted limitations of current laboratory models of urban flooding as regards pointwise velocity measurements;
- mentioned the challenges related to laboratory experiments of dual drainage as well as to the representation of frictional resistance;
- clarified the experience-based nature of the recommendations by Chanson (2004);
- detailed the uncertainties in the inflow discharges;
- included a reference to full scale laboratory tests of urban flooding,
- used more specific wording at several instances.

Also, all technical comments by the Referees have been implemented.

In the following, we present a point-by-point response to all the comments raised by the Referees. To a great extent, the present document consists in an updated version of our responses provided during the discussion phase, in which we also describe how the revisions have indeed been incorporated in the manuscript. Note that all line numbers in our responses refer to the *revised* version of the manuscript.

Best regards,

X. Li, Corresponding author

**Referee 1**

Corresponding changes are highlighted in yellow in the revised manuscript.

**General comment**

> The authors present an interesting paper on scale effects in the context of the modelling of urban flooding using geometrically distorted models, based on the reanalysis of existing lab data. I am not aware of any similar analysis done in this context, and the paper will be of significant interest to those undertaking such studies. The paper is well written and clear and I would certainly recommend publication. I have made some suggestions for potential improvements below.

Thank you for the positive feedback and useful comments on how to improve the manuscript.

**Specific comments**

> 1. The abstract (L17) contains a comparison of uncertainties from scale effects to those from hydrological data. This analysis does not seem to be in the paper or conclusions. The abstract should not be contain information that is not in the full paper; hence this should be removed or added to the paper. Please also note the typo on the last line.

As suggested by the Referee, this sentence has been deleted in the revised manuscript.

> 2. Introduction (2$^{nd}$ paragraph) – The authors point out the difficulty in obtaining velocity data in field conditions with which to validate flood models. However point velocity data is also uncommon for lab studies of urban flooding (for example due to the low flow depths and the difficulty in installing equipment such as ADV probes). Indeed the studies the authors use is based on flow and depth data. Some studies which provide velocity data in a lab setting have recently come available (e.g. Martins et al 2018, WRR) and the authors may at some point in the paper wish to consider the benefits of velocity + depth data for model validation or scale model studies.

We concur with the Referee in acknowledging the difficulty of conducting point velocity measurements in experimental models of urban flooding at the district level.

Therefore, in the revised manuscript,

- we discuss explicitly this issue in the Introduction (P2, L9-14);
- we also emphasize the need for more pointwise velocity measurements as a perspective in the Conclusion (P11, L21-23).

We understand that the Editor recommends inclusion of these items in a Recommendation section. However, we find a bit odd to create a new section to cover this issue (since a section Recommendation does not exist in the present version of the manuscript), and we find more effective to just include dedicated paragraphs in the Introduction and at the end of the Conclusion. If really needed, we are ready to reconsider this.

3. P2 L16 – it would benefit the introduction if the authors could be a bit more specific regarding what they mean by 'artefacts'. Perhaps with a specific example of how these might arise in scaled experiments, what is the physical cause etc.

According to the *Oxford English Dictionary*, an "artefact" is "*A spurious result, effect, or finding in a scientific experiment or investigation, esp. one created by the experimental technique or procedure itself*."

Here, what we mean by *artefacts* are deviations between up-scaled model measurements and real-world prototype observations due to governing non-dimensional parameters (i.e. force ratios) which are not identical between the model and its prototype (Heller, 2011). This may include alteration of the flow regime (transition vs. complete turbulent), of the relative importance of frictional resistance or of 2D and 3D flow structures.

This is now clarified in the revised manuscript (P2, L26-29).

4. P2 L25 – Chanson gives some specific recommendations, however it would be good if the basis for these recommendations was briefly mentioned.

Although the recommendations by Chanson (2004) look very reasonable, no specific background information is referred to by Chanson (2004) to support the stated recommendations (scale factors below 25 to 50, distortion ratio below 5 to 10), which are mostly based on experience (personal communication with H. Chanson). Nonetheless, in the revised manuscript, we have now clarified the field of application for which these recommendations were formulated (P2, L35) and we refer explicitly to "experience-based" (P2, L37; P4, L24).

5. P3-P4 The authors provide a good review of physical models and scaling issues in the simulation of surface urban flood flows. The authors should briefly note that studies of 'dual drainage' (surface and sewer flows) also exists however the scaling issues are somehow complicated by the combination of pressurized pipe and surface flow. This generally posed some additional limitations on these studies (either associated with the hydraulic conditions which can be simulated, or the minimum size of the model).

The Referee highlights that scale effects in "dual drainage" models is even more intricate due to the combination of pressurized pipe flow and surface flow. Since the body of the present manuscript (especially the datasets) focuses solely on surface flow in a network of streets (not dual drainage), we propose to mention dual drainage as a perspective in the Conclusion of the revised manuscript.

Therefore, next to the sentence stating that "More controlling parameters should also be considered, such as the bottom slope …", we have now added a paragraph on the importance of dual drainage for urban flooding and the associated experimental modelling challenges (P11, L18-21):

*"Moreover, in real-world urban flooding, the urban drainage system may also have a substantial influence. Laboratory modelling of dual drainage becomes even more intricate due to the combination of pressurized and surface flow (e.g. Rubinato et al., 2017). This poses additional constraints on the design of the scale models and leads to extra experimental challenges which would deserve further research."*

Again, since dual drainage is somehow "out of the scope" of the analysis we conduct based on the two specific datasets, we find it difficult to include dual drainage in the Discussion section. This may be reconsidered if needed.

95

> 6. P5 – For both datasets the authors do not discuss the quantification of the inflow discharge. It seems to be assumed that the error within the quantification of this discharge is negligible. Some justification for this assumption should be given.

We agree that information on the inflow discharge uncertainties were missing in the original manuscript. They have
100    now been added in Section 3.1 of the revised manuscript (P5, L23 and L36).

Regarding the consequences of these uncertainties on the inflow discharge, we consider that they are lumped in the uncertainty estimate derived from the analysis of the test repetitions, as detailed in Supplements 2 and 3. This is now stated explicitly in Section 4.1 of the revised manuscript (P7, L23-L24):

*"Uncertainties arising from inaccuracies in the inflow discharge are also lumped into this uncertainty estimate."*

105

> 7. The treatment of frictional resistance within scaled models is an interesting topic which the authors mention only briefly within the discussion. Given such models are generally constructed from smooth materials can authors comment further on the representation of roughness within scaled models and any implications? Why is friction expected to be underestimated in more distorted models?

110    As stressed by the referee, the representation of frictional resistance in scale models of urban flooding is particularly challenging. In this respect, we have clarified the following aspects in the revised manuscript.

- When we write that "friction is expected to be underestimated in more distorted models", we mean "underestimated" compared to the case of an undistorted model with similar bed material (not necessarily compared to the prototype!, as clarified now ). This is true because the more distorted the model, the higher
115    the Reynolds number and the lower the relative roughness ($k_{s,m} / R_{H,m}$). Both of these effects contribute to lower the friction factor, as confirmed by the experimental results in Figure S23. This is now stated explicitly on P8, L30-L31.

- In the original manuscript, we hardly explained that, even without distortion, frictional resistance may not be properly represented in the scale model, particularly when smooth material is used to construct the
120    bottom and the walls of the model. There are two competing effects (lower Reynolds number in the model compared to the prototype; but also lower relative roughness) which, in general, hamper a definite prediction on whether frictional resistance is over- or under-estimated compared to the prototype. This is now stated explicitly in the revised version of the Introduction (P2, L39 - P3, L2).

125 | 8. P9 L11 – Can the authors comment on why they expect third effect to be probably dominant? Perhaps this should be phrased 'The third of these effects…'

Indeed, "the third type of effects" being dominant is not well supported at this stage. This would require additional detailed measurements and/or interpretation based on computational modelling. Our point is that, at the present stage, we may not exclude that this third effect has a substantial influence; but we are unable to confirm it based

130 on the existing data. Therefore, in the revised manuscript, the sentence has been rephrased as follows (P9, L31):

*"It is possible that the third of these effects has a substantial influence on the flow. However, the available datasets ..."*

**Technical comments**

135 | 1. P2 L2 'than ever observed in the past' should be removed (too many points in one sentence)

This has been removed in the revised manuscript in P.2 L3.

2. P3 L6 'replace 'were' with 'have been'

The word 'were' has been replaced by 'have been' in revised manuscript in P.3 L21.

140

3. P3 L11 – Seems to have a divide symbol?

Indeed, the divide symbol '÷' has now been replaced by the more standard symbol '-'. The change has been implemented throughout the manuscript (8 occurrences: P.3 L28-29; P.4 L38; Table 1).

145 | 4. P4 L34 - Replace 'it was never' with 'has not to date been'

The wording 'it was never' has been replaced by 'has not to date been' in the revised manuscript in P.5 L9.

**Referee 2**

150      Corresponding changes are highlighted in green in the revised manuscript.

**General comment**

The manuscript presents an interesting study about the influence of horizontal/vertical distortions of lab hydraulic models – to the best of my knowledge this is a novel investigation in the topic. The study is based on two existing datasets that were generated with other objectives in mind and, for this reason, the results are (somewhat) not fully conclusive, as the authors mention at the later sections of the manuscript. Nevertheless, in my opinion, the results are relevant to improve urban drainage lab modelling, and perhaps more important, the study identifies the need of new lab experiments to improve the understanding of urban drainage systems hydraulics. The manuscript is well written and clear. My main concern is the lack of consideration of uncertainties in hydraulic measurements and their influence in the conclusion of the study. Below I suggest a few points that the authors may consider to improve the quality of the manuscript.

Thank you for these supportive comments and insightful suggestions to improve the manuscript.

We concur with Referee 2 regarding the importance of considering uncertainties in hydraulic measurements. However, we believe that these uncertainties are already carefully reported and discussed in the manuscript. Particularly, Section 4.1 is entirely devoted to the evaluation of the experimental uncertainties, with additional details provided in Supplements 2, 3, 4 and 5, as briefly summarized hereafter.

- To estimate the uncertainties affecting the experimentally measured *water depths*, we account both for the instrument accuracy and for the repeatability of the tests (flow variability). The latter is detailed in Supplements 3 and 4. The water depth profiles provided in Supplement 4 also display the corresponding uncertainties (~ 2 mm). Importantly, Figs. 3 and 5 compare the influence of the experimental uncertainties and the effect of model distortion on the upscaled water depths. See also P. 8 L9-11 and L18-19: "… despite the uncertainties in the measurements, the differences between the model runs increase very consistently as the model distortion increases …", as well as L1-2 on p. 9: " … the differences arising from the change in model distortion greatly exceed the estimated experimental uncertainty."

- The uncertainties on estimated *outflow discharges* are also reported in Section 4.1 and details on the repeatability of the measurements are presented in Supplement 5 (see also L27-28 on P. 7: "comparing the repeated measurements demonstrates an excellent repeatability of the outflow discharge estimates"). The standard deviation of estimated outflow discharges is displayed in Fig. S21, which supports the comments formulated in Section 4.3.

In addition, all our main results are presented with the corresponding uncertainties (Fig. 3, Supplements 6, 7 and 8).

**Specific comments**

Abstract – The meaning of 'artefacts' should be presented, otherwise it is too vague.

We have now detailed what we mean by "artefacts" in the Introduction of the revised manuscript (P2, L26-29).
185  Additionally, since the word "artefacts" is also used in the Abstract, we have now mentioned examples of such artefacts in the revised version of the Abstract (P1, L13-14).

P1, L34: the authors refer to aerials images as a source for reference data. Although this is in theory valid, in many urban flooding situations the events are of short duration making capture of aerial data (e.g. from satellite and un-
190  or manned aircraft sensors) challenging.

We agree with the Referee about the challenges associated to the use of aerial imagery to document (short duration) urban flooding. This is actually what we meant already by "… aerial imagery are available; but they remain uncertain and insufficient …". To be more specific in the revised manuscript, we explicitly mention the issue of time resolution: "insufficient (e.g. inadequate time resolution)" (P1, L36).

195

P2, L6: the authors may also want to refer other recent lab experiments that were conducted at the real scale (e.g. Moy de Vitry et al., 2017 – ESSD). These, of course, do have other challenges than those investigated in the present manuscript, but should also be discussed in the context of urban drainage hydraulic lab experiments.

In the revised manuscript, we refer to the study by Moy de Vitry et al. (2017), which uses a full scale urban flooding
200  model to explore the potential of unconventional data (video data and computer vision) for monitoring urban flooding (P2, L7-9).

P2, L11: the authors refer to 'both scales' at this point of the manuscript without yet having defined it. I assume the authors are referring to horizontal and vertical scales. Please clarify.

205  The wording "both scales" does not refer to the horizontal and vertical scales; but it refers to the "prototype" and the "model". This has now been clarified in the revised manuscript (P2, L20).

P3, L35: 'Using a distorted model…'. It seems that the sentence should start with something like 'By using a distorted…'.

210  Indeed, this has been rephrased as 'By using a distorted model' in the revised manuscript in P.4 L11.

> P5, L19: '… at least four to six…'. I believe the authors can be more specific here. The mentioned experiment as conducted in 2017, so it should not be difficult to know exactly how many runs were carried out!

When writing "'… at least four to six experimental runs …", we did not mean that we do not know the exact number of runs; but simply that the number of runs depends on the considered layout of obstacles, as detailed in Table 3:

- 4 runs are available for the layout "Narrow streets …"
- 6 runs are available for the layout "Intermediate streets …"
- 6 runs are available for the layout "Wide streets …"

This has now been made clearer in the revised manuscript (P5, L35): "For each of these layouts, between four and six experimental runs (Table 3) were conducted with various steady inflow discharges …".

> P5, L22: 'In contrast, …', I cannot understand to what (sentence or idea) this sentence is contrasting to. Please consider rephrasing it.

We actually mean "In contrast with the dataset of Araud (2012) …", the dataset of Velickovic et al. (2017) does not include discharge data. This sentence has been clarified in the revised manuscript (P6, L1).

> P7, L16: '…may be …' the use of this words does not sound very scientific. Please consider a different construction, e.g. 'is between'

As suggested by the Referee, this sentence has been rephrased as follows (P7, L36): "… the standard deviation remains below 2 to 6 mm (depending on the experimental run)".

> 2. 9: P8, L12-17: this seems to belong to the methodology section. It is definitely not a result. Please consider moving these sentences to another section.

These sentences do present results, because upscaled water depths (not the original experimental observations) are obtained by applying the methodology developed in the present study.

> P8, L26: '…almost…'. This is not very scientific word. How many? What %?

The words "almost" and "most" used in the Section 4.3 refer to a general trend; but we agree with the Referee that it deserves being quantified. Hence, we use now a much more specific wording:

- P9, L12-13: "… *in the range of ± 10 % in 11 streets out of 14 (and in the range of ± 12 % in 13 streets out of 14, with a maximum change of + 18 % in just a single street)*"

- P9, L8-9: "*... the variation of the outflow discharge partition with d shows a consistent monotonous trend (increasing, constant or decreasing) in all outlet streets*"

245

> Figure 6a: does the figure report V (V = SUM Q) or Q? this should be made clear. Also, if it is Q, the variation can be included (error bars?).

Figure 6a presents the outflow discharge in each street, as described in Section 4.3. The variations of outflow discharge are presented in Fig. S21 in the form of error bar.

250

> 2. 12: P9, L9: 'Aspect ratio'. Should this be defined? Is it similar to hydraulic radius? Please explicitly define the meaning of 'aspect ratio'

The aspect ratio is defined as the ratio between the water depth and the street width. This definition was indeed missing in the original manuscript and it has now been added in Section 5.1 of the revised manuscript, where it is
255    first used (P9, L29).

[revised manuscript text omitted]

---

## Author Response (AR3)

**COVER LETTER**

Dear Editor,

Thanks a lot for your prompt handling of our resubmitted manuscript.

We fully understand the concern of ensuring maximum consistency between the scope of the paper and that of the journal Therefore, we have now incorporated a number of changes to make clearer the links between our study and the main themes of HESS.

Mainly, we have included extra keywords and sentences, using wordings such as: "small-scale flow processes", "urban hydrological models", "low impact development", "parametrized in urban hydrology models". We have also added 4 new references to papers published recently in HESS.

We hope that this will improve the discoverability of the paper for colleagues involved in flood risk modelling.

Best regards,

X. Li, Corresponding author

[revised manuscript text omitted]